# Attitude to Co-Administration of Influenza and COVID-19 Vaccines among Pregnant Women Exploring the Health Action Process Approach Model

**DOI:** 10.3390/vaccines12050470

**Published:** 2024-04-28

**Authors:** Alessandra Fallucca, Palmira Immordino, Patrizia Ferro, Luca Mazzeo, Sefora Petta, Antonio Maiorana, Marianna Maranto, Alessandra Casuccio, Vincenzo Restivo

**Affiliations:** 1Department of Health Promotion, Mother and Child Care, Internal Medicine and Medical Specialities, University of Palermo, 90127 Palermo, Italy; palmira.immordino@unipa.it (P.I.); patrizia.ferro@unipa.it (P.F.); luca.mazzeo@unipa.it (L.M.); sefora.petta@gmail.com (S.P.); alessandra.casuccio@unipa.it (A.C.); 2HCU Obstetrics and Gynaecology, ARNAS Civico Di Cristina—Benfratelli Hospital, 90127 Palermo, Italy; antonio.maiorana@arnascivico.it (A.M.); marianna.maranto@arnascivico.it (M.M.); 3School of Medicine, University Kore of Enna, 94100 Enna, Italy; vincenzo.restivo@unikore.it

**Keywords:** pregnant women, co-administration of vaccines, influenza vaccination, COVID-19 vaccination, health action process approach model

## Abstract

Respiratory tract diseases caused by influenza virus and SARS-CoV-2 can represent a serious threat to the health of pregnant women. Immunological remodulation for fetus tolerance and physiological changes in the gestational chamber expose both mother and child to fearful complications and a high risk of hospitalization. Vaccines to protect pregnant women from influenza and COVID-19 are strongly recommended and vaccine co-administration could be advantageous to increase coverage of both vaccines. The attitude to accept both vaccines is affected by several factors: social, cultural, and cognitive-behavioral. In Palermo, Italy, during the 2021–2022 influenza season, a cross-sectional study was conducted to evaluate pregnant women’s intention to adhere to co-administration of influenza and COVID-19 vaccines. The determinants of vaccination attitude were investigated through the administration of a questionnaire and the Health Action Process Approach theory was adopted to explore the cognitive behavioral aspects. Overall, 120 pregnant women were enrolled; mean age 32 years, 98.2% (*n* = 118) of Italian nationality and 25.2% (*n* = 30) with obstetric or pathological conditions of pregnancy at risk. Factors significantly associated with the attitude to co-administration of influenza and COVID-19 vaccines among pregnant women were: high level of education (OR = 13.96; *p* < 0.001), positive outcome expectations (OR = 2.84; *p* < 0.001), and self-efficacy (OR = 3.1; *p* < 0.001). Effective strategies to promote the co-administration of the influenza vaccine and the COVID-19 vaccine should be based on the communication of the benefits and positive outcomes of vaccine co-administration and on the adequate information of pregnant women.

## 1. Introduction

Respiratory infections caused by the influenza virus and SARS-CoV-2, if acquired during pregnancy, increase the risks of spontaneous abortion, preterm birth, cesarean section, stillbirth and fetal anomalies, including organogenesis defects and intrauterine growth restriction [1]. There are several pathophysiological and immunological mechanisms involved in the increased risk of these respiratory infections for pregnant women and their fetuses, including: increased intra-abdominal pressure, elevation of the diaphragm, remodulation of the immune system and intensified activation of cytokines [2,3,4,5].

The influenza virus represents a threat to the health of pregnant women due to high epidemiological impact and related high burden of disease. The World Health Organization recommends influenza vaccination for all fragile subjects, including pregnant women, to prevent serious outcomes and complications [6]. Inactivated influenza vaccines have an excellent safety and efficacy profile and, according to the Italian National Vaccine Prevention Plan (PNVP 2023–2025), they can be administered at any time during gestation with the benefits of the vaccine extending to both the pregnant women and newborns [7]. There is evidence that the influenza vaccine is effective both in reducing the risk of hospitalization for acute respiratory disease in pregnant women and in conferring effective protection to the fetus and also to the newborn after birth, up to six months of life [8,9].

Similarly to the Influenza virus, SARS-CoV-2, which has more recently emerged, is a largely ubiquitous virus responsible for numerous cases of respiratory disease with possible severe outcomes for subjects with predisposing clinical conditions. There are still gaps in understanding the real impact of the virus on pregnant women. In the early stages of the pandemic, research on COVID-19 complications in pregnant women was limited [10]. Over time, however, the association between gestation and increased disease severity for COVID-19 became evident. Several hospitalizations in intensive care units with invasive ventilation and numerous cases of premature membrane rupture have been reported in SARS-CoV-2 positive pregnant women [11,12].

In a short time after the launch of the COVID-19 vaccination, the collection of safety data and the growing evidence of substantial harm of COVID-19 to pregnant women has led many health institutions to unequivocally recommend vaccination of pregnant women [13,14]. 

Despite strong support from the main health authorities, vaccination hesitancy among pregnant women remains high [15,16]. Refusal to vaccinate despite the availability of vaccines has often been associated with a lack of perception of vaccine safety and efficacy. The decision-making process on vaccination acceptance can be explored by cognitive behavioral models that investigate outcome expectations and perception of risk and safety, such as the Health Action Process Approach theory (HAPA), the Health Belief Model (HBM) or the transtheoretical model of change [17,18,19]. 

Vaccine decision-making is also influenced by social, demographic, medical conditions and morbidity factors that could be determinants in the vaccination among pregnant women [20,21,22]. Strategies that make vaccination of pregnant women a priority are needed. Vaccine co-administration, defined as the simultaneous administration of two vaccines in a single session, could represent a useful and advantageous preventive practice to reduce costs, facilitate vaccination logistics and allow full adherence to recommended vaccinations [23]. In detail, the co-administration of influenza and COVID-19 vaccines could be advantageous because these two respiratory viruses co-circulate during the cold season and the vaccination campaigns are conducted in the same period of the year.

With the aim of investigating the factors that influence vaccine uptake in pregnant women, a cross-sectional survey was conducted on the attitudes and behaviors of pregnant women in the city of Palermo, Italy, towards vaccination during pregnancy against influenza and COVID-19.

## 2. Materials and Methods

### 2.1. Study Design

A cross-sectional study was conducted to investigate the attitude of pregnant women to receive co-administration of influenza and COVID-19 vaccines. The investigation took place during the 2021–2022 influenza season, at the ARNAS Civico Di Cristina—Benfratelli Hospital in Palermo, Italy, a large hospital with more than 800 beds, of which 30 are for the gynecology ward, representing the reference center for Sicilian maternity care, counting around two thousand births in a year. The study was approved by the ethical committee Palermo 1 on 18 December 2020. The pregnant women were interviewed with the help of a questionnaire and the answers were collected anonymously, in accordance with articles 13 and 14 of the GDPR 2016/679 EU for the protection of natural persons with respect to the processing of personal date. Participation in the study was offered to pregnant women who went to the hospital’s outpatient department for scheduled visits during the gestation period. Adult pregnant women, aged 18 years and older, Italian and non-Italian, with a good knowledge of the Italian language, and who had given informed consent to participate in the study, were included. Pregnant women who did not meet these requirements were excluded from the study. 

### 2.2. Search Team

The recruitment of pregnant women was conducted by a team of several health workers: health assistants and doctors from the School of Specialization in Hygiene and Preventive Medicine of the University of Palermo, experts in the field of vaccination, and gynecologists from the ARNAS Civico Di Cristina—Benfratelli Hospital of Palermo, involved in the outpatient management of pregnant women.

### 2.3. The Questionnaire

A questionnaire consisting of two different sections was administered to interview pregnant women. The first section concerned personal data and socio-demographic context, such as age, nationality, residence, marital status, occupation, educational level. According to the literature, the variable “educational level” was categorized as follows: “Low” for primary school and secondary school qualification, “Medium” for high school diploma and “High” for a degree or higher qualification [24]. The variable “Occupation” was arbitrarily categorized based on self-reported data by the women interviewed: the possible answer options to describe employment status were “Employed”, “Unemployed” and “Housewife”. The first section also investigated aspects related to pregnancy with items about the week of gestation, risk factors and attitude to anti-influenza and COVID-19 vaccination.

The second section consisted of the cognitive model adopted in this study, the HAPA theory. Developed in the 1990s by Ralf Schwarzer, the HAPA theory suggests that the initiation and maintenance of a health behavior are part of a two-phase process, consisting of an initial motivation phase, in which the individual develops the intention to adopt a precautionary measure, and of a second volitional-voluntary phase, in which the individual chooses to implement the health behavior and plans the actions to be implemented [17]. The HAPA theory was adopted with eight items, investigating four different domains of the model: “risk perception” of adverse events of respiratory infections contracted during pregnancy, expectations of a “positive outcome” and “negative outcome” related to co-administration of vaccination during the gestational period, and “self-efficacy”, the ability to complete the preventive action one intends to take, in this case, to receive two vaccines simultaneously. The available response options, according to a five points Likert scale, ranged from 1 = “I strongly disagree” to 5 = “I strongly agree”. For each respondent, the scores relating to the two items of the same domain were added (range score 2–10 per domain) and the median value was calculated. For further information about the questionnaire, see the Appendix A.

### 2.4. Statistical Analysis

The data relating to the answers of the pregnant women were transferred from paper formats to computer files. A database was used to record the data using Excel—Office 2021. All collected data were analyzed using Stata/SE 14.2 statistical software (Copyright 1985–2015, StataCorp LLC, 4905 Lake-way Drive, College Station, TX, USA. Revised 29 January 2018). 

The normality of the distribution of the quantitative variables was evaluated by the Skewness and Kurtosis’s test. Normally distributed quantitative variables were summarized as mean (standard deviation) and not-normally distributed variables as median (interquartile range). For the qualitative variables, absolute and relative frequencies were calculated. The association of normally and non-normally distributed quantitative variables with co-administration of influenza and COVID-19 vaccines was assessed by the Student’s *t*-test and the Wilcoxon and Mann-Whitney’s tests, respectively, while for the qualitative variables the Chi2 test was used. A logistic regression analysis was performed to evaluate the factors associated with vaccination attitude against influenza and COVID-19. A multivariable backward stepwise logistic regression model was used to analyze the covariates associated at univariable analysis with a *p* value equal or lower than 0.05. Variables related to HAPA domains were assumed as confounders a priori. For all analyses, a *p* value of 0.05 was assumed to indicate significance (two-tailed).

## 3. Results

Overall, 120 pregnant women were enrolled and answered the questionnaire with the mean age of 32 years (28–35). Most of the women interviewed were of Italian nationality (98.3%; *n* = 118) and resided in the city of Palermo (54.2%; *n* = 65). About marital status, 76.7% were married (*n* = 92) and 13.3% lived with their partner (*n* = 16). With regard to the level of education, about a third of the respondents declared to have a university degree (36.7%, *n* = 44) and about a third to have a high school diploma (36.7%, *n* = 44). Almost half of pregnant women were employed (57.5%; *n* = 69) and 58.3% of the respondents were in their first pregnancy (*n* = 70). About a quarter of women had a high-risk pregnancy (25.2%, *n* = 30); the risk condition was related to obstetric conditions for 43.3% (n = 13) and to maternal pathologies for 56.7% (*n* = 17). The main source of information on vaccines was the gynecologist (36.7%; *n* = 44), followed by the general practitioner (35%; *n* = 42). Furthermore, investigating the intention of pregnant women to receive the two vaccines simultaneously, it was found that approximately two-thirds (66.7%; *n* = 80) would not adhere to vaccine co-administration against influenza and COVID-19 (Table 1).

The comparison between pregnant women who were willing to adhere to vaccine co-administration and those who were not showed that women who were willing to receive vaccines were slightly older in age (33 years vs. 31 years; *p* = 0.028), more frequently university graduates (65% vs. 22.5%; *p* = 0. 001) and employed (75.5% vs. 36.2%; *p* < 0.001). Furthermore, women who potentially adhere to co-administration about values for HAPA domains, had a higher perception of positive vaccination outcomes (8 vs. 6; *p* < 0.001) and a higher likelihood of successful completion of the preventive action of receiving the two vaccines simultaneously (8 vs. 6; *p* < 0.001) (Table 1).

The HAPA construct analysis showed that the majority of pregnant women had a low perception of the risk of abortion related to influenza virus or SARS-CoV-2 infection, 33.3% (*n* = 40) were ‘undecided’ and 27.5% (*n* = 33) were ‘disagree’, respectively; similarly, 40% (*n* = 48) indicated ‘strongly disagree’ with the increased risk of caesarean section caused by respiratory infections (Table 2). Regarding expectations of a positive outcome, the majority of women ‘agreed’ that co-administration of the vaccines could have reduced the risk of hospitalization (43.3%; *n* = 52) and that the vaccines could also have protected their children after birth in the first few months of life (55.8%; *n* = 67). The women interviewed feared the negative outcomes of vaccines, “agreeing” that vaccines against COVID-19 and influenza could have cause systemic side effects (62.5%; *n* = 75) and local effects at the injection site (67.5%; *n* = 81) (Table 2). Furthermore, the majority of women believed they did not have enough information to decide to receive two vaccines co-administered, 41.7% (*n* = 50) were “disagree” and 8.3% were “strongly disagree”, respectively; but, at the same time, almost two thirds of pregnant women agreed that the contrary opinion of family and friends could not influence the eventual decision to receive the two vaccines (61.7%; *n* = 74) (Table 2).

Regarding multivariable analysis, factors significantly associated with attitudes to influenza and COVID-19 vaccine co-administration among pregnant women were: high level of education, degree or higher, (OR = 10.7; 95% CI = 1.16–98.02), positive outcome expectations (HAPA domain), regarding the perception of a reduction in disease-related complications and the protection of children in the first months of life (OR = 2.06; 95% CI = 1.22–3.49), and self-efficacy (HAPA domain), i.e., adequate information regarding vaccinations and ability to adhere to preventive practices even without a favorable opinion from family members and friends (OR = 2.45; 95% CI = 1.59–4.52) (Table 3).

## 4. Discussion

Infections during pregnancy represent a current public health problem as there are multiple populations at risk of complications: the pregnant women, the fetus and the future unborn child [1,11]. In detail, respiratory tract infections, such as influenza and COVID-19, could be responsible for a high disease burden related to the anatomical and physiological changes characteristic of pregnancy [3,4]. The need to explore the attitude of pregnant women to receive vaccinations against respiratory viruses, such as influenza virus and SARS-CoV-2, and to evaluate the factors associated with the vaccine decision-making led to the conduction of this study.

In Italy, in accordance with the objectives of national health planning and with the specific objectives of the influenza immunization program, influenza vaccination must be offered actively and free of charge to subjects who, due to personal conditions, have a greater risk of complications. According to the Italian National Vaccine Prevention Plan, pregnant women are among the priority categories, together with the population over 65 years [7]. For the high-risk population, the optimal influenza vaccination coverage target is set at 95% [25].

Every year, at the end of the influenza vaccination campaign, the Italian Ministry of Health promptly provides data on vaccination coverage among the Italian population, distinguishing between the coverage of the general population and the over 65 population [26]. To date, however, no data have been made available on influenza vaccination coverage among the pregnant women. The lack of a vaccination registry for pregnant women represents an obstacle for timely monitoring of the progress of vaccination coverage. Furthermore, the lack of data related to vaccination rates does not facilitate understanding the real impact of respiratory infections on the clinical conditions of fragile patients like pregnant women.

In the United States, a system for monitoring the vaccination status of pregnant women has been developed with the collaboration of the CDC’s Immunization Safety Office and multiple integrated health organizations. During the influenza season, monthly estimates of influenza vaccination coverage are provided for pregnant women based on electronic health record data [27]. This kind of vaccination monitoring system allows the real-time evaluation of the vaccination rate of a category particularly susceptible to complications, such as pregnant women. The implementation of local computerized vaccination registers, interoperable with the national one, represent one of the fundamental strategies for prevention envisaged by the Italian National Prevention Plan [28]. In detail, the timely monitoring of vaccinations during pregnancy would be particularly useful as it would allow vaccination promotion interventions to be promptly implemented to protect both pregnant women and future unborn children.

Similar to those reported for influenza vaccination, reports of COVID-19 vaccine administrations were also transmitted stratified by age group but not by risk category, during the pandemic as well as during the booster dose administration phases [29]. Although the recommendations have always been aimed at prioritizing COVID-19 vaccination to the most fragile categories with a high risk of hospitalization, the data on the doses administered have never been detailed according to risk targets [30,31].

Some studies have provided estimates on the vaccination rate among pregnant women in Italy [32,33,34]. To date, however, there are few studies in the literature that have aimed to estimate the simultaneous adherence to influenza and COVID-19 vaccinations in this specific risk category of women. Furthermore, this is one of the first studies that investigated factors that could facilitate uptake of vaccines co-administration among Italian pregnant women.

The results of this study showed that one in three pregnant women would comply with the co-administration of the influenza and COVID-19 vaccination. This adherence percentage appears far from the target of 95% vaccination coverage recommended for pregnant women [25]. The US pregnancy immunization monitoring system reported higher rates of compliance with both vaccinations during the 2021–2022 season: 44% for the influenza vaccine and 71% for the COVID-19 vaccine [27]. However, a cross-sectional study conducted in Italy revealed a lower rate (23%) of acceptance of simultaneous vaccination for influenza and COVID-19 among the general population [35]. The higher acceptance declared by Italian pregnant women, compared to the Italian general population, could probably be attributed to a greater need for protection and to the motivation of protecting their child as well as themselves.

This survey demonstrated that a factor significantly associated with the attitude to receive both vaccines was the expectation of positive outcomes from co-administration of the vaccine, an item investigated using the constructs of the HAPA model. The perception of the benefit of vaccines in conferring protection to the mother, from possible complications and hospitalizations, and to the future unborn child, from infection in the first months of life, could have a decisive role in the vaccination decision-making process and, in particular, in the acceptance of a co-administration vaccination. This finding is confirmed by a US study that explored the intention to receive two vaccines in the same session, in which it emerged that the perceived benefits could motivate something more than the threat of infection and its consequences [36].

The analysis of the HAPA cognitive behavioral model showed another determining factor of the attitude towards co-administration of influenza and COVID-19 vaccination related to self-efficacy. Women who believed they had both adequate information on vaccinations and autonomy in the decision to adhere to preventive practices even with the contrary opinion of family and friends were more inclined to receive the two vaccines. Self-efficacy refers to an individuals’ belief to exercise control over the adoption of a behavior even in the face of obstacles [17,37]. A study that explored several cognitive behavioral theories related to influenza vaccination adherence confirmed the decisive role of self-efficacy that could effectively mediate vaccine perceptions and behavioral intentions [38]. Self-efficacy could also be described as an individual ability that can increase with practice and exercise. Prevention education could improve pregnant women’s self-efficacy in adhering to vaccination practices [39]. In the Italian study of adherence to influenza and COVID-19 vaccine co-administration among the general population, it was found that acceptance of co-administration was significantly higher among those who had actively sought information about vaccines than among those who had been passively exposed to the information [35]. The promotion of active and correct information about prevention could actually positively influence vaccine-related decision-making among pregnant women and lead to greater adherence to co-administration.

One of the main factors associated with the intention to receive simultaneous influenza and COVID-19 vaccines among pregnant women was a higher level of education. This factor has been investigated in many studies on adherence to preventive immunization practices [40,41]. It has already been identified as a strong positive predictor of adherence to the anti-COVID-19 vaccination in pregnant women, but also as a facilitator in the uptake of other vaccinations recommended during pregnancy, such as against pertussis and influenza [33]. Frequently, in fact, the higher educational qualification was associated with greater knowledge about vaccines and higher awareness of the benefits associated with vaccination [42]. More educated people probably have access to more accurate information on health and vaccination as they consult more accredited sources of information. The role of information sources in the vaccination field could therefore be crucial [43]. Most pregnant women in our sample reported that they were informed about vaccinations by a gynecologist or general practitioner. It is imperative that health care providers, who assist women during the gestation period, actively raise awareness of the risks related to respiratory infections in pregnancy and adequately inform pregnant women about the benefits related to the coadministration of influenza and COVID-19 vaccines.

The limited sample size and self-reported nature of the data on vaccination adherence could raise doubts about the representativeness of pregnant women of the Palermo area. Furthermore, the survey was conducted anonymously in accordance with articles 13 and 14 of the GDPR 2016/679 EU for the protection of natural persons with respect to the processing of personal data. These conditions did not allow verification of the vaccination status of pregnant women. The attitude of pregnant women towards the vaccine against Diphtheria-Tetanus-Pertussis has not been investigated as the vaccination co-administration for adults currently involves a maximum of two administrations. Despite these limitations, this study was one of the first of its kind to explore pregnant women’s attitudes towards influenza and COVID-19 vaccination, and to provide useful data to effectively promote co-administration of vaccination during pregnancy. Effective strategies to promote the co-administration of influenza and the COVID-19 vaccines should be based on the communication of the benefits and positive outcomes of vaccine co-administration and on the adequate information of pregnant women.

## 5. Conclusions

The intention to accept vaccine co-administration during pregnancy could be related to several socio-cultural and behavioral determinants, and the topic deserves to be further investigated. The effective promotion of vaccinations against respiratory viruses, such as influenza virus and SARS-CoV-2, could be focused on counselling about the expectation of positive outcomes of vaccine co-administration and on the adequate information of pregnant women about vaccinations by gynecologists and other healthcare workers. Furthermore, it could be necessary to adopt a computerized system, standardized across the territory, which records data on vaccinations during pregnancy, to monitor the progress of coverage and to plan future vaccination interventions aimed at the target category.

## Figures and Tables

**Table 1 vaccines-12-00470-t001:** Characteristics of pregnant women and differences between with or without co-administration.

		Total Respondents *n* (%)	Co-Administration *n* (%)	Without Co-Administration *n* (%)	*p* Value
		120	40	80	
Age					
	Median value	32 (28–35)	33 (29.6–35)	31 (27.5–34)	0.028
Nationality					
	Italian	118 (98.3%)	39 (97.5%)	79 (98.7%)	0.600
	Foreign	2 (1.7%)	1 (2.5%)	1 (1.3%)	
Residence					
	Palermo city	65 (54.2%)	20 (50.0%)	45 (56.0)	0.078
	Province of Palermo	39 (32.5%)	12 (30%)	27 (33.7%)	
	Other Sicilian cities	8 (6.7%)	2 (5.0%)	6 (7.5%)	
	Other Italian cities	8 (6.7%)	6 (15%)	2 (2.5%)	
Educational level					
	Low	32 (26.7%)	3 (7.5%)	29 (36.2%)	<0.001
	Medium	44 (36.7%)	11 (27.5%)	33 (41.3%)	
	High	44 (36.7%)	26 (65%)	18 (22.5%)	
Occupation					
	Employed	69 (57.5%)	30 (75%)	39 (48.7%)	0.006
	Unemployed	13 (10.8%)	5 (12.5%)	8 (10.0%)	
	Housewife	38 (31.7%)	5 (12.5%)	33 (41.3%)	
Marital status					
	Single	4 (3.3%)	3 (7.5%)	1 (30.1%)	0.289
	Engaged	5 (4.2%)	2 (5.0%)	3 (32.3%)	
	Married	92 (76.6%)	29 (72.5%)	63 (26.8%)	
	Divorced	3 (2.5%)	0	3 (9.7%)	
	Cohabitant	16 (13.3%)	6 (15%)	10 (12.5%)	
Week of pregnancy					
	Median value	37 (36-38)	37 (36-39)	37 (36-39)	0.104
N° of children					
	0	70 (58.3%)	26 (65%)	44 (55%)	0.830
	1	31 (25.8%)	9 (22.5%)	22 (25.5%)	
	2	14 (11.7%)	4 (10.0%)	10 (12.5%)	
	3	4 (3.3%)	1 (2.5%)	3 (3.8%)	
	5	1 (0.8%)		1 (1.3%)	
Risk pregnancy					
	No risk	90 (75.0%)	27 (63.55)	63 (78.8%)	
	Clinical risk	17 (14.2%)	6 (15.0%)	11 (13.7%)	0.230
	Obstetric risk	13 (10.8%)	7 (17.5%)	6 (7.5%)	
Source of information					
	Gynecologist	44 (36.7%)	15 (37.5%)	29 (36.3%)	
	HCW F.C. *^1^	3 (2.5%)	2 (5%)	1 (1.3%)	
	General practitioner	42 (35%)	12 (30%)	30 (37.5%)	
	Pediatrician	11 (9.2%)	1 (2.5%)	10 (12.5%)	0.201
	HCW V.C. *^2^	3 (2.5%)	2 (5%)	1 (1.3%)	
	Friends	1 (0.8%)		1 (1.3%)	
	Family	1 (0.8%)		1 (1.3%)	
	TV/Media	4 (3.3%)	1 (2.5%)	3 (3.8%)	
	Web/Internet	2 (1.7%)	1 (2.5%)	1 (1.3%)	
	Other	9 (7.5%)	6 (15%)	3 (3.8%)	
HAPA domains *^3^					
	Risk perception	5 (4–6.5)	4.5 (4–7)	5 (4–6%)	<0.001
	Positive Outcome	7 (6–8)	8 (7–5)	6 (5–7%)	
	Negative Outcome	8 (6–8)	8 (6–8)	8 (6–8%)	
	Self-efficacy	6 (6–8)	8 (7–8.5)	6 (5–7%)	

*^1^ HCW F.C. = Healthcare worker of Family Clinic; *^2^ HCW V.C. = Healthcare Worker of the Vaccination Center; *^3^ mean value of Likert scale answers; 1 = strongly disagree; 2 = disagree; 3 = undecided; 4 = agree; 5 = strongly agree. Range score for item = 2–10.

**Table 2 vaccines-12-00470-t002:** Characteristics of health action process approach items for pregnant women.

Risk Perception	Strongly Agree	Agree	Undecided	Disagree	Strongly Disagree
SARS-CoV-2 and influenza infection could increase the risk of miscarriage	3 (2.5%)	29 (24.2%)	40 (33.3%)	33 (27.5%)	15 (12.5%)
SARS-CoV-2 and influenza infection could increase the risk of resorting to cesarean delivery	0	19 (15.8%)	34 (28.3%)	48 (40%)	19 (18.3%)
**Positive Outcomes**	**Strongly Agree**	**Agree**	**Undecided**	**Disagree**	**Strongly Disagree**
Co-administration of COVID-19 and influenza vaccines could reduce the risk of being hospitalized for complications	9 (7.5%)	52 (43.3%)	27 (22.5%)	28 (23.3%)	4 (3.3%)
Co-administration of COVID-19 and influenza vaccines could protect my baby in few first months of life	8 (6.7%)	67 (55.8%)	24 (20%)	17 (14.2%)	4 (3.3%)
**Negative Outcomes**	** *Strongly Agree* **	** *Agree* **	** *Undecided* **	** *Disagree* **	** *Strongly Disagree* **
Co-administration of COVID-19 and influenza vaccines could lead to the same frequency of side effects, such as fever or headache, as if the individual vaccines were given separately	4 (3.3%)	75 (62.5%)	30 (25%)	11 (9.2%)	0
Co-administration of COVID-19 and influenza vaccines could lead to the same frequency of side effects, such as pain, redness and swelling in the arms despite of given the individual vaccines	5 (4.2%)	81 (67.5%)	23 (19.2%)	10 (8.3%)	1 (0.8%)
**Self-Efficacy**	** *Strongly Agree* **	** *Agree* **	** *Undecided* **	** *Disagree* **	** *Strongly Disagree* **
I am confident to have enough information about co-administration of COVID-19 and influenza vaccines to make the decision to get them	4 (3.3%)	29 (24.2%)	27 (22.5%)	50 (41.7%)	10 (8.3%)
I am sure to have co-administration even though my family/friends disagree	17 (14.2%)	74 (61.7%)	10 (8.3%)	18 (15.0%)	1 (0.8%)

**Table 3 vaccines-12-00470-t003:** Univariable and multivariable analysis of factors associated with attitude to vaccines co-administration.

		Crude OR	p(z)	Adjusted OR	p(z)
Age					
	Per unit increase	1.09 (1.00–1.18)	0.031	1.03 (0.89–1.18)	0.655
Residence					
	Palermo city	Ref		Ref	
	Province of Palermo	1 (0.42–2.36)	1.000	0.64 (0.17–2.34)	0.502
	Other Sicilian cities	0.75 (0.13–4.04)	0.738	1.26 (0.08–19.5)	0.868
	Other Italian cities	6.74 (1.15–30.4)	0.026	1.52 (0.14–15.69)	0.722
Marital status					
	Single	Ref			
	Engaged	0.222 (0.01–3.97)	0.307		
	Married	0.153 (0.01–1.54)	0.111		
	Divorced	1			
	Cohabitant	0.20 (0.016–2.38)	0.203		
Educational level					
	Low	Ref			
	Medium	3.22 (0.81–12.6)	0.094	3.34 (0.50–22.07)	0.209
	High	13.96 (3.68–52.89)	<0.001	10.7 (1.16–98.02)	0.036
Risk pregnancy					
	No risk	1			
	Clinical risk	1.27 (0.42–3.79)	0.665		
	Obstetrician risk	2.72 (0.83–8.85)	0.096		
Occupation					
	Employed	Ref			
	Unemployed	0.81 (0.24–2.73)	0.738	1.64 (0.28–9.63)	0.579
	Housewife	0.19 (0.06–0.56)	0.003	2.24 (0.36–13.69)	0.381
Source of information					
	Gynecologist	Ref			
	HCW F.C. *^1^	3.86 (0.32–46.1)	0.285		
	General practitioner	0.77 (0.31–1.92)	0.582		
	Pediatrician	0.19 (0.02–1.65)	0.124		
	HCW V.C. *^2^	3.86 (0.32–46.1)	0.285		
	Friends	1			
	Family	1			
	TV/Media	0.64 (0.61–6.74)	0.714		
	Web/Internet	1.9 (0.84–17.6)	0.649		
HAPA domains *^3^					
	Risk perception	0.90 (0.72–1.12)	0.357	0.76 (0.52–1.13)	0.180
	Positive outcome	2.84 (1.85–4.36)	<0.001	2.06 (1.21–3.49)	0.007
	Negative outcome	1.01(0.76–1.35)	0.922	1.21 (0.75–1.93)	0.436
	Self-efficacy	3.10 (2.01–4.76)	<0.001	2.45 (1.59–4.52)	<0.001

*^1^ HCW F.C. = Healthcare worker of Family Clinic; *^2^ HCW V.C. = Healthcare Worker of the Vaccination Center; *^3^ mean value of Likert scale answers; 1 = strongly disagree; 2 = disagree; 3 = undecided; 4 = agree; 5 = strongly agree. Range score for item = 2–10.

## Data Availability

Data will be available after to motivated request to corresponding author.

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
