# Peer review of "Attitude to Co-Administration of Influenza and COVID-19 Vaccines among Pregnant Women Exploring the Health Action Process Approach Model"

_vaccines, 2024, doi:10.3390/vaccines12050470_

Round 1

Reviewer 1 Report

Comments and Suggestions for Authors

Dear Editor

Dear Authors

Important work in terms of prevention among pregnant women.

What can be improved:

The introduction contains more information about knowledge gaps and the impact of the study on reality.

The methodology should be divided into sections to ensure transparency, and it should be done in accordance with the journal's guidelines: there should be clear information about the research team, research tool, inclusion and exclusion criteria, ethical considerations, statistical analysis, etc.

There are places in the discussion where there is no reference to literature.

Author Response

Reviewer 1

Dear Editor

Dear Authors

Important work in terms of prevention among pregnant women.

What can be improved:

Q: The introduction contains more information about knowledge gaps and the impact of the study on reality.

A: The introduction of the manuscript has been modified following your suggestion. Some "not essential" periods to adequately introduce the study conducted have been removed or summarized. Thank you for your note.

Q: The methodology should be divided into sections to ensure transparency, and it should be done in accordance with the journal's guidelines: there should be clear information about the research team, research tool, inclusion and exclusion criteria, ethical considerations, statistical analysis, etc.

A: Thank you for your suggestion. The “Materials and methods” section has been extensively modified to clearly explain how the investigation was conducted. Furthermore, the questionnaire was uploaded as "Supplementary material", both in Italian and in English, to allow researchers interested in the topic to access the survey instrument used.

Q: There are places in the discussion where there is no reference to literature.

Thanks for this comment. We have added bibliographic references in some periods of the Discussion section.

“Infections during pregnancy represent a current public health problem as there are multiple populations at risk of complications: the pregnant women, the fetus and the future unborn child [1],[11]. In detail, respiratory tract infections, such as influenza and COVID-19,could be responsible for a high disease burden related to the anatomical and physiological changes characteristic of pregnancy [3,4].”

“According to the Italian National Vaccine Prevention Plan, pregnant women are among the priority categories, together with the population over 65 years [7].  For the high-risk population, the optimal influenza vaccination coverage target is set at 95% [25].”

Reviewer 2 Report

Comments and Suggestions for Authors

Thank you for the opportunity to evaluate this paper for publication in Vaccines. The topic is interesting; however, the current presentation of the manuscript is not suitable for publication, and there are several weaknesses that require more careful attention.

Firstly, the introduction discusses the negative effect of respiratory infections during pregnancy, but then focuses only on two of the many possible respiratory infections during pregnancy. The text should start by describing only these two among others. A major limitation is the small sample size. While it is true that this is an observational study and therefore sample size computation is not needed, the authors should compute the size of the overall population they should observe to reach a pre-specified level of precision of prevalence estimates of their outcomes (based on the literature). If this is excessively distant, the research may need to recruit additional subjects.

Furthermore, the statistical analysis requires a more careful review. The construction of multivariate models would benefit from a stepwise approach, defining a significance level at the univariate level to be included in the final model. This would help to obtain more precise estimates, avoiding the wide confidence intervals of the odds ratios (a symptom of imprecision due to the small sample size). Additionally, the large size of some components of certain variables makes inferential analyses less credible.
Perhaps adding the entire questionnaire used as supplementary material would help to understand some additional data, as it is not very clear how the research instrument is constructed.
Lastly, it is not clear why there is a paragraph titled "6. Patent."
In a nutshell, I would like to read a revised version of the paper in which results and discussion are redefined based on a more coherent approach.

Author Response

Reviewer 2

Thank you for the opportunity to evaluate this paper for publication in Vaccines. The topic is interesting; however, the current presentation of the manuscript is not suitable for publication, and there are several weaknesses that require more careful attention.

Q: Firstly, the introduction discusses the negative effect of respiratory infections during pregnancy, but then focuses only on two of the many possible respiratory infections during pregnancy. The text should start by describing only these two among others.

A: The introduction of the manuscript has been modified by presenting, from the first line, the two respiratory infections under investigation: seasonal influenza and SARS-CoV-2 infection. These are two common respiratory infections for which there are vaccines that are recommended during pregnancy and that can be administered during pregnancy. Following your suggestion, we specified in the "limitations" paragraph that our study did not investigate the aptitude of pregnant women to receive another recommended vaccine: the vaccine against Diphtheria - Tetanus - Pertussis.

Q: A major limitation is the small sample size. While it is true that this is an observational study and therefore sample size computation is not needed, the authors should compute the size of the overall population they should observe to reach a pre-specified level of precision of prevalence estimates of their outcomes (based on the literature). If this is excessively distant, the research may need to recruit additional subjects.

A: Thanks for your suggestions. During the study design phase, the simple sampling formula was used to calculate the sample size. Considering a precision=0.05 and a prevalence=0.10 (10%) the sample size was N=138 subjects. We recruited a total of 120 pregnant women. The study sample size is smaller than the estimated sample size but the difference is not much. The "limited sample size", together with the other limitations of the study, was reported in the manuscript, at the end of the "Discussions".

Q: Furthermore, the statistical analysis requires a more careful review. The construction of multivariate models would benefit from a stepwise approach, defining a significance level at the univariate level to be included in the final model. This would help to obtain more precise estimates, avoiding the wide confidence intervals of the odds ratios (a symptom of imprecision due to the small sample size). Additionally, the large size of some components of certain variables makes inferential analyses less credible.

A: Thank you for your suggestion. The logistic analyses were performed according to the stepwise approach. It was probably not clear because just significant variables of the univariate analysis were reported in the Table 3. We have updated the "Methods - Statistical analyses" section and Table 3. Furthermore, following the suggestion of another peer reviewer, we also performed a logistic analysis with a cut-off of 20%. The statistically significant results (p<0.05) remained unchanged, demonstrating the strength of the analyses. The multivariable model suggested by the reviewer reduces the number of observations to 114. It is preferable to use the multivariable model we proposed because it refers to 120 observations, i.e. the entire sample.

Q: Perhaps adding the entire questionnaire used as supplementary material would help to understand some additional data, as it is not very clear how the research instrument is constructed.

  1. Thank you for your suggestion. The “Materials and methods” section has been extensively modified to clearly explain how the investigation was conducted. Furthermore, the questionnaire was uploaded as "Supplementary material", both in Italian and in English, to allow researchers interested in the topic to access the survey instrument used.

Q: Lastly, it is not clear why there is a paragraph titled "6. Patent."

A: We have removed the "Patent" paragraph from the manuscript. Thank you for the suggestion.

Q: In a nutshell, I would like to read a revised version of the paper in which results and discussion are redefined based on a more coherent approach.

  1. We have made several changes to the manuscript. In detail, introductions, materials and methods, tables and limits have been modified to present the study more clearly and coherently. We hope that both the changes made and the arguments discussed through the "point to point" answers will meet your expectations.

Reviewer 3 Report

Comments and Suggestions for Authors

Manuscript ID: vaccines-2952206

Title: Attitude to co-administration of Influenza and Covid-19 vaccines among pregnant women exploring the Health Action Process Approach model

COMMENTS TO THE AUTHORS

General comments

The authors investigated the attitudes and behaviors of pregnant women in the city of Palermo, Italy, towards vaccination during pregnancy against influenza and COVID-19. This is important work since vaccine co-administration could be a useful and advantageous preventive measure to reduce costs, facilitate vaccination processes and increase adherence to recommended vaccinations. However, the authors should redo the multivariate analysis (MVA) to explore whether adding other covariates not initially statistically significant by themselves at 5% in bivariate analyses to see whether they improve the model (see details below).

Specific comments for revision:

a)      Major

    • The authors reported that they forced the HAPA variables in the MVA and kept only covariates associated with the dependent variable at the 5% significance level. I think the authors need to redo the MVA by also including other covariates associated with the dependent variable at a significance level higher than 5%. I suggest a cut-off of 20% and doing a stepwise method to keep those that stay significant at 5% along with the HAPA variables in the MVA. Variables not initially significant by themselves at 5% in bivariate analyses may become statistically significant when included with other covariates in a MVA regression.
    • Page 6, lines 195-202: please report 95% confidence intervals instead of p values.
    • Table 3 title: please replace “associated to” with “associated with”.
    • Table 3: please get rid of columns p(z). Confidence intervals suffice.
    • Table 3: Tables should stand alone. Please add a footnote with a brief description of the HAPA variables and their ranges/categories.

b)      Minor

    • Page 3, line 106: typo “committee”.
    • Page 3, line 109: typo “nationality”.
    • Page 4, line 153: please delete “to” in “answered to the questionnaire”.
    • Page 5, line 177: please rewrite how the subjects reported their Likert scale answers.  Also, please only keep the percentages or Ns (preferably percentages). Maybe change to something like this: “…33.3% were undecided while 27.5% disagreed. Similarly, 40% strongly disagreed with the increased risk of caesarean section…”.
Comments on the Quality of English Language

There are several typos in the manuscript and some sentences need to be revised/rewritten!

Author Response

Reviewer 3

General comments

The authors investigated the attitudes and behaviors of pregnant women in the city of Palermo, Italy, towards vaccination during pregnancy against influenza and COVID-19. This is important work since vaccine co-administration could be a useful and advantageous preventive measure to reduce costs, facilitate vaccination processes and increase adherence to recommended vaccinations. However, the authors should redo the multivariate analysis (MVA) to explore whether adding other covariates not initially statistically significant by themselves at 5% in bivariate analyses to see whether they improve the model (see details below).

Specific comments for revision:

Q: Major

The authors reported that they forced the HAPA variables in the MVA and kept only covariates associated with the dependent variable at the 5% significance level. I think the authors need to redo the MVA by also including other covariates associated with the dependent variable at a significance level higher than 5%. I suggest a cut-off of 20% and doing a stepwise method to keep those that stay significant at 5% along with the HAPA variables in the MVA. Variables not initially significant by themselves at 5% in bivariate analyses may become statistically significant when included with other covariates in a MVA regression.

A: Thank you for your suggestion. The logistic analyses were performed according to the stepwise approach. It was probably not clear because just significant variables of the univariate analysis were reported in the Table 3. We have updated the "Methods - Statistical analyses" section and Table 3. Following your suggestion, we also performed a logistic analysis with a cut-off of 20%. The statistically significant results (p<0.05) remained unchanged, demonstrating the strength of the analyses. Furthermore, the suggested multivariable model reduces the number of observations to 114. It is preferable to use the multivariable model proposed by us because it refers to 120 observations, i.e. the entire sample.

Q: Page 6, lines 195-202: please report 95% confidence intervals instead of p values.

A: According to your suggestion, 95% confidence intervals for the results of the logistic analysis are reported in the text.

Q: Table 3 title: please replace “associated to” with “associated with”.

A: Thank you for the correction.

Q: Table 3: please get rid of columns p(z). Confidence intervals suffice.

A: Thank you for your suggestion. Reporting confidence intervals instead of p values in the text can facilitate a more precise interpretation of the data. In the tables, however, we prefer to report both to give the opportunity to interested readers to have both confidence interval and p-value.

Q: Table 3: Tables should stand alone. Please add a footnote with a brief description of the HAPA variables and their ranges/categories.

A: We have added a footnote to explain HAPA after Table 1 and Table 3. “*mean value of Likert scale answers; 1= strongly disagree; 2= disagree; 3= undecided; 4= agree; 5= strongly agree. Range score for item = 2 – 10”

Q:  Minor

Page 3, line 106: typo “committee”.

A: Thank you for your suggestion. "The study was approved by the ethical committee Palermo 1 on 18/12/2020. "

Q: Page 3, line 109: typo “nationality”.

A: Thank you for the suggestion. “The first section concerned personal data and socio-demographic context, such as age, nationality, residence, marital status, occupation, educational level”

Q: Page 4, line 153: please delete “to” in “answered to the questionnaire”.

A: Thank you for the suggestion we modified the sentence as following “Overall, 120 pregnant women were enrolled and answered the questionnaire with the mean age of 32 years (28-35).”

Q: Page 5, line 177: please rewrite how the subjects reported their Likert scale answers.  Also, please only keep the percentages or Ns (preferably percentages). Maybe change to something like this: “…33.3% were undecided while 27.5% disagreed. Similarly, 40% strongly disagreed with the increased risk of caesarean section…”.

A: Thank you for your suggestion. The Likert scale answers of the pregnant women interviewed are reported in the manuscript, immediately after Table 1. "The HAPA construct analysis showed that the majority of pregnant women had a low perception of the risk of abortion related to influenza virus or SARS-CoV-2 infection, respectively 33.3% (n=40) were 'undecided' and 27.5% (n=33) were 'disagree'; similarly, 40% (n=48) were 'strongly disagree' with the increased risk of caesarean section caused by respiratory infections...".

We have preferred to present both percentages and numbers, in brackets, in the Results section. Having a not very large sample of pregnant women, the percentage alone may not allow an adequate understanding of the data.

Please see the attachment for analysis.

Round 2

Reviewer 2 Report

Comments and Suggestions for Authors

Authors fully addressed the reviewers’ comments.